computational biology/pattern recognition

dimensionality reduction, outlier detection, high-dimensional data, genomics

**Author for correspondence:**
Omar Shetta
e-mail: os10g13@soton.ac.uk

# Robust subspace methods for outlier detection in genomic data circumvents the curse of dimensionality

## Omar Shetta and Mahesan Niranjan

Electronics and Computer Science, University of Southampton, Southampton SO17 1BJ, UK

OS, 0000-0001-9634-3419

The application of machine learning to inference problems in biology is dominated by supervised learning problems of regression and classification, and unsupervised learning problems of clustering and variants of low-dimensional projections for visualization. A class of problems that have not gained much attention is detecting outliers in datasets, arising from reasons such as gross experimental, reporting or labelling errors. These could also be small parts of a dataset that are functionally distinct from the majority of a population. Outlier data are often identified by considering the probability density of normal data and comparing data likelihoods against some threshold. This classical approach suffers from the curse of dimensionality, which is a serious problem with omics data which are often found in very high dimensions. We develop an outlier detection method based on structured low-rank approximation methods. The objective function includes a regularizer based on neighbourhood information captured in the graph Laplacian. Results on publicly available genomic data show that our method robustly detects outliers whereas a density-based method fails even at moderate dimensions. Moreover, we show that our method has better clustering and visualization performance on the recovered low-dimensional projection when compared with popular dimensionality reduction techniques.

## 1. Introduction

The problem of cancer classification, and identifying clinically relevant tumour subgroups are aided by monitoring gene expression [1,2]. Moreover, gene expression profiling is one of the key approaches used to find potential biomarkers and therapeutic targets for distinct cancer types [3]. However, these large datasets are often affected by outliers. In common language, outliers are a small fraction of samples that deviate

considerably from other samples in the population. Outliers can arise from errors in the experimental procedure or can be samples that are functionally different from the majority of the population. In the former case, they are discarded to prevent them from affecting downstream statistical analysis [4], and in the latter case, outliers can be further analysed to find that they belong to a rare cell type or to a functionally distinct group of cells [5]. Therefore, machine learning techniques that are robust to outliers are of great interest, as they will be able to compute models that are not affected by abnormalities, and will be able to detect outliers. As an example, robust regression has been applied recently with much success in [6,7]. This work shows that in some cases concentration of proteins in yeast cells could be predicted from mRNA abundance in addition to sequence-derived features. Extracted outliers from their model were recognized as being subject to post-translational modifications. Another approach for outlier and anomaly detection is to fit a probability density function on the data. Methods such as Gaussian mixture models and kernel density estimation have been used by [8,9] to detect novelties in different applications. However, more recently, Aggarwal [10] has discussed that using a mixture of Gaussian components can overfit a cluster of outliers. This configuration happens frequently in real settings where the outliers have a high similarity with each other. Using a single Gaussian component works surprisingly well in practice for outlier detection tasks [10]. However, probability density fitting methods will break down if applied directly to gene expression datasets because they suffer from high dimensionality, as their number of features (genes) is much greater when compared with their number of samples. The problem with high-dimensional datasets is that when the number of features increases the volume of the space increases in such a rapid manner that the available samples are not sufficient to get statistically significant results. By reducing the dimensionality of the data and keeping the same number of samples, it will be possible to apply statistical techniques to extract useful information. This will solve the issues caused by the curse of dimensionality. Therefore, it is of great interest to reduce the dimensionality of gene expression datasets without losing too much useful biological information. A widely used dimensionality reduction technique is principal component analysis (PCA), which showed its importance during the past years in data analysis, especially on high-dimensional transcriptomic datasets [11]. PCA seeks to find a low-dimensional subspace that has the smallest least-squares reconstruction error [12]. However, it is known to be heavily affected by outliers in the data, even in the presence of one outlier [13,14]. This is mainly because the least-squares error that is minimized in the PCA objective function has a quadratic term which will amplify the errors produced by the outliers in the data. This motivated many researchers over the past years to find formulations of PCA that are robust to outliers. However, many robust PCA algorithms suffer from two main drawbacks: computational intractability and degradation of performance when the dimensionality of the data increases [15]. A robust PCA method that considers these two drawbacks is outlier pursuit (OP) which is introduced by Xu *et al*. [15]. OP considers the problem of recovering the column space of the uncorrupted points and the index of the outlier points that are present in the data by minimizing a convex objective function. This convexity makes the problem solvable by simple optimization methods that can find a global minimum of the objective function. On the contrary, the state-of-the-art robust PCA methods are non-convex and optimization methods will converge to local minima. However, OP does not take into account the inherent manifold structure of the data; this is also a known drawback of standard PCA. This gives misleading or unsatisfactory results when it comes to highly nonlinear datasets such as transcriptomic and proteomic datasets where features and samples can have complex relationships with each other.

In this paper, we focus on gene expression data which will naturally have a highly complex structure. To solve this issue, we will introduce graph-regularized outlier pursuit (GOP), where we add a graph regularization term to the objective function of OP, with the aim to find a low-dimensional representation that respects the intrinsic geometric structure that the data lives in.

This paper is organized as follows: In §2, we will introduce both algorithms for OP, graph-regularized OP (GOP) and the Gaussian density estimation method which is used as a benchmark for outlier detection. In §3, we will show that the robust subspace methods will not be affected by the high dimensionality of the datasets compared to fitting a Gaussian density. We also show that GOP gives better outlier detection performance than outlier pursuit and the Gaussian density method on publicly available gene expression datasets. Furthermore, we will show that GOP has a more discriminative low-dimensional space when it comes to separating outliers from the main samples when compared with popular dimensionality reduction techniques. In §4, we will highlight the importance of our method compared to similar methods in outlier detection applications. Finally, we end with concluding remarks in §5.

# 2. Material and methods

## 2.1. Outlier pursuit

We will consider from this point onwards that our input data matrix $M \in \mathbb{R}^{m \times n}$ has $n$ samples arranged in columns with each sample having $m$ features, $M = [M_1, M_2, ..., M_n]$. Where $M_i \in \mathbb{R}^m$ denotes the $i$th column of matrix $M$. We consider the outliers to be fully corrupted columns. The OP objective is to decompose the data matrix $M$ as $M = L + C$. Where $L$ is a low-rank matrix and $C$ is a column-sparse matrix which has a small fraction of its columns that are non-zero. This method is modelling outlier samples as the non-zero columns in $C$, where they are considered the corrupted points of the data matrix. Moreover, OP models the uncorrupted column space that needs to be recovered as the column space of the low-rank matrix $L$. OP introduced by [15] seeks to minimize the following function

$$\min_{L,C} \|L\|_* + \lambda \|C\|_{1,2} \quad \text{subject to: } M = L + C, \tag{2.1}$$

where $\|L\|_*$ is the nuclear norm of $L$ and it is defined as the sum of its singular values. $\|C\|_{1,2}$ is the sum of the $l_2$ norm of the columns of $C$. $\lambda$ is a regularization parameter which needs to be tuned; this will be addressed in §2.6. Problem 2.1 is convex, thus it is efficiently solved using first-order optimization methods. The algorithm to solve this problem is given in algorithm 1,

---

**Algorithm 1.** Accelerated proximal gradient **(outlier pursuit)**

---

**input:** $M \in \mathbb{R}^{m \times n}$ ,$\lambda$, $\delta = 10^{-5}$, $\eta = 0.9$

 $\mu_0 = 0.99 \|M\|_F$.

 (i) choose initial value of $C^0$, $C^{-1}$, $L^0$, $L^{-1} \in \mathbb{R}^{m \times n}$; $t^0, t^{-1} \leftarrow 1$; $\bar{\mu} \leftarrow \delta \mu_0$

 (ii) **repeat the following until convergence**

 (iii) $Y_L^k = L^k + \dfrac{t^{k-1} - 1}{t^k}(L^k - L^{k-1})$ ; $Y_C^k = C^k + \dfrac{t^{k-1} - 1}{t^k}(C^k - C^{k-1})$;

 (iv) $(U, S, V) = \text{svd}\left(Y_L^k + \dfrac{1}{2}(Y_L^k + Y_C^k - M)\right)$;

 (v) $L^{k+1} = U \xi_{\frac{\mu_k}{2}}(S) V^T$;

 (vi) $C^{k+1} = \zeta_{\frac{\mu^k \lambda}{2}}\left(Y_C^k + \dfrac{1}{2}(Y_L^k + Y_C^k - M)\right)$

 (vii) $\mu^{k+1} = \max\left(\eta \mu^k, \bar{\mu}\right)$

 (viii) $t^{k+1} = \dfrac{1 + \sqrt{4t_k^2 + 1}}{2}$; $k {++}$

**output:** $\hat{L} = L^k$, $\hat{C} = C^k$ when $k$ is last iteration.

---

where $L^k$ stands for $L$ at iteration $k$, and $\|M\|_F$ is the Frobenius norm of matrix $M$ defined by $\|M\|_F = \sqrt{\sum_{i=1}^n \|M_i\|_2^2}$. In step (v), $\xi_\epsilon(S)$ is the singular value soft-thresholding operator which acts on the diagonal elements of $S$. If $|S_{ii}| \leq \epsilon$, then $S_{ii}$ is set to zero, otherwise it is shrunk by $\epsilon$, i.e $S_{ii} := S_{ii} - \epsilon$. Furthermore in step (vi), $\zeta_\epsilon(C)$ is the soft-thresholding operator on columns of C, such that if $\|C_i\|_2 \leq \epsilon$ ($C_i$ is the $i$th column of C) set $C_i = 0$, otherwise set $C_i := C_i - \epsilon \cdot C_i / \|C_i\|_2$.

## 2.2. Outlier pursuit algorithm

Problem 2.1 can be solved using a first-order optimization method called accelerated proximal gradient (APG) method, which benefits from an optimal convergence rate of $O(1/k^2)$ [16] (where $k$ is the number

of iterations). APG method is the accelerated version of the more general proximal gradient method which has a convergence rate of $O(1/k)$. The improvement in convergence rate is achieved by the momentum step devised by [16], in steps (iii) and (viii) of algorithm 1. For the interested reader, the references that validate this algorithm are [16–18].

## 2.3. Graph-regularized outlier pursuit

Graph-regularized outlier pursuit incorporates in its objective function the intrinsic manifold information of the data in the form of a graph. The graph which has nodes corresponding to samples, is constructed by first finding the $K$ nearest neighbours of each sample. Then for each sample, we weight the edges to its $K$ neighbours through the Gaussian kernel function $W_{ij} = \exp^{-\|M_i - M_j\|_2^2/2\sigma^2}$. All other points that are not in the $K$ nearest neighbours of the sample are weighted as zero. During our work, we choose $K$ as being smaller than or equal to the expected number of outliers present in the data. The matrix that incorporates this information is the affinity matrix $W \in \mathbb{R}^{n \times n}$. Then the graph Laplacian matrix $\Phi \in \mathbb{R}^{n \times n}$ is defined by $\Phi = D - W$, where $D$ is a diagonal matrix where each entry on its diagonal is the row sum of the corresponding row in $W$, $D_{ii} = \sum_j W_{ij}$. The objective function of graph-regularized outlier pursuit is as follows:

$$\min_{L,C} \|L\|_* + \lambda \|C\|_{1,2} + \alpha \operatorname{tr}(L\Phi L^T) \quad \text{subject to: } M = L + C. \tag{2.2}$$

GOP seeks to find the best linear embedding of the data that is robust to outliers, while enhancing the embedding through the graph regularizer. It will achieve this by pushing points closer together in the low-dimensional space if they have high affinity $W_{ij}$ in the original input space. This will preserve the intrinsic nonlinear structure present in the data while finding the best robust low-rank approximation to the data matrix. To best interpret the function of the graph regularization term $\operatorname{tr}(L\Phi L^T)$, we can rewrite it in the following way:

$$\operatorname{tr}(L\Phi L^T) = \sum_{i=1}^{n} L_i^T L_i D_{ii} - \sum_{i,j=1}^{n} L_i^T L_j W_{ij}$$

$$= \frac{1}{2} \sum_{i,j=1}^{n} \|L_i - L_j\|_2^2 W_{ij}.$$

The graph regularization term can be better interpreted now as $\frac{1}{2}\sum_{i,j=1}^{n} \|L_i - L_j\|_2^2 W_{ij}$. This function will impose structure in the recovered low-rank matrix $L$, in the sense that if two points have high affinity in the original input space the distance of the corresponding columns in $L$ needs to be small. Moreover, the graph regularization term would enhance separability of outliers in the $L$ matrix, when this separable structure is found by the affinity matrix $W$. Furthermore, we need to emphasize that problem (2.2) is a convex problem, and it can be solved using alternation direction method of multipliers (ADMM) [19]. The algorithm is shown in the next section. We also need to emphasize that GOP is used in this work to detect outlier samples. However, this method can further be used to detect outlier genes if only all the matrices in problem (2.2) are transposed.

## 2.4. GOP algorithm

To solve GOP using ADMM, we need to introduce an auxiliary variable so that we can divide the objective function into three separate blocks. We rewrite the GOP objective function as follows:

$$\min_{L,C,Q} \|L\|_* + \lambda \|C\|_{1,2} + \alpha \operatorname{tr}(Q\Phi Q^T) \quad \text{subject to: } M = L + C, \quad L = Q, \tag{2.3}$$

where $Q$ is an auxiliary variable. Now, we can define the augmented Lagrangian function of (2.3)

$$\mathcal{L}(L, C, Q, Z_1, Z_2) = \|L\|_* + \lambda \|C\|_{1,2} + \alpha \operatorname{tr}(Q\Phi Q^T)$$

$$+ \langle Z_1, M - L - C \rangle + \frac{p_1}{2}\|M - L - C\|_F^2$$

$$+ \langle Z_2, Q - L \rangle + \frac{p_2}{2}\|Q - L\|_F^2,$$

where $\langle \cdot, \cdot \rangle$ denotes the Frobenius inner product of two matrices, if $\langle X, Y \rangle$ then it is defined as $\operatorname{tr}(X^T Y)$. Here, we need to minimize the augmented Lagrangian with respect to each of the five variables

**Algorithm 2.** Alternating direction method of multipliers (**graph-regularized outlier pursuit**)

**input:** $M \in \mathbb{R}^{m \times n}$ ,$\lambda$,$\alpha$, $\Phi$, $p_1 = 1$, $p_2 = 1$

(i) initialize $L^0$, $C^0$, $Q^0$ to random matrices.

(ii) $Z_1^0 = M - L^0 - C^0$ and $Z_2^0 = Q^0 - L^0$.

(iii) **repeat following until convergence**

(iv) $\quad L^{k+1} = \underset{L}{\text{argmin}}\mathcal{L}\,(L, C^k, Q^k, Z_1^k, Z_2^k)$

(v) $\quad C^{k+1} = \underset{C}{\text{argmin}}\mathcal{L}\,(L^{k+1}, C, Q^k, Z_1^k, Z_2^k)$

(vi) $\quad Q^{k+1} = \underset{Q}{\text{argmin}}\mathcal{L}\,(L^{k+1}, C^{k+1}, Q, Z_1^k, Z_2^k)$

(vii) $\quad Z_1^{k+1} = Z_1^k + p_1(M - L^{k+1} - C^{k+1})$

(viii) $\quad Z_2^{k+1} = Z_2^k + p_2(Q^{k+1} - L^{k+1})$

**output:** $\hat{L} = L^k$, $\hat{C} = C^k$ when $k$ is last iteration.

sequentially. The general form of the ADMM algorithm to solve GOP is shown in algorithm 2, where $Z_1^k$ and $Z_2^k$ are the Lagrange multipliers and $k$ is the iteration index.

Steps (iv), (v) and (vi) have closed form solutions and they are derived in the electronic supplementary material. We should emphasize that there exists a method that uses a closely related method to GOP which is robust PCA on graphs (RPCAG) by [20]. The main difference between these two methods is the model of sparsity of the reconstruction matrix $C$. In [20], the reconstruction matrix is modelled by a $l_1$ norm which induces overall sparsity of the whole matrix. This model is more suitable to images, which is what they demonstrate their results on. In the case of gene expression data, a model of column sparsity fits more efficiently the model of outliers. The main algorithmic difference between GOP and Shahid's robust PCA model is in the update of the $C$ matrix, step (v) of algorithm 2. We show in the electronic supplementary material the enhanced outlier detection performance of GOP compared to RPCAG, proving that a column sparsity constraint models more accurately the outliers in gene expression datasets.

## 2.5. Detecting outliers using OP and GOP

The objective of both OP and GOP is to decompose the input data matrix into a low-rank plus a column-sparse matrix. In the real dataset case, we need to consider that noise will be present and the recovered low-rank matrix $\hat{L}$ and column-sparse matrix $\hat{C}$ will be corrupted by noise. This will result in a $\hat{C}$ matrix that is not strictly column sparse, but will have high $l_2$ norm for columns that are considered outliers [15] Therefore, for both OP and GOP, we use two methods to detect the outliers:

(1) $\hat{C}$ **method**: Rank $l_2$ norms of columns of $\hat{C}$ in descending order and choose outliers to be the points with $l_2$ norm higher than a threshold.

(2) $\hat{L}$ **method**: First, find the singular value decomposition (SVD) of the recovered $\hat{L}$ matrix, $\hat{L} = U \Sigma V^T$. Then find the low-dimensional embedding $Z$ by projecting $\hat{L}$ onto its column space $U$, $Z = U^T\hat{L}$. Finally, perform $k$-means clustering onto $Z$ to fit two clusters. The minority cluster is chosen to be the outliers.

For the first method, we can only choose a small fraction of highest points as being the outlier. This method suffers from the drawback that choosing a fraction of outliers needs prior knowledge of the domain. However, for the second method, $k$-means clustering will decide which points correspond to which cluster by giving cluster labels; therefore, giving a cut-off between outlier and main samples without deciding an outlier fraction *a priori*.

The F-score of two cluster $k$-means on the low-dimensional embedding $Z$ is used to quantify the performance of outlier detection of the $\hat{L}$ method. The F-score is a measure of accuracy and will give a higher score if the outlier and main samples are better separated in the low-dimensional embedding. The F-score is defined by

$$\text{F-score} = 2\,\frac{\text{precision} \cdot \text{recall}}{\text{precision} + \text{recall}},$$

where, precision = true positives/(true positives + false positives) and recall = true positives/positives. Here, the positives class is defined as the known outliers present in the datasets that we use. Datasets are introduced in §2.9.

## 2.6. Parameter setting for OP and GOP

Tuning parameters for unsupervised problems such as GOP and OP is more challenging than supervised problems where the true labels are available. The presence of labels for supervised problems makes it possible to measure the accuracy of detection, which can be used as a metric to choose optimal regularization parameters. In the case of outlier detection algorithms, using the knowledge of true outliers in the tuning process would not be practical as users will not know the true outliers beforehand. In the case of GOP and OP, the two factors that are affected from the regularization parameters are the rank of $\hat{L}$ and the number of outliers detected. Therefore, we can only use both of them to tune the regularization parameters $\lambda$ and $\alpha$. The outliers during the tuning process will be detected using the $\hat{L}$ method explained in §2.5. The tuning process consists of solving the problem for each value of $\lambda$ in a specific range, and looking for stable regions of the rank of $\hat{L}$. We then refine the $\lambda$ search space to the stable region and record the number of outliers. A suitable value of $\lambda$ needs to be chosen in such a way that the number of detected outliers are less than or equal to an expected fraction of outliers. From our studies, we suggest to expect a fraction of outliers that is less than 25% of the data. Moreover, we found practically that the number of outliers and the rank of $\hat{L}$ are not affected by the value of $\alpha$, thus we choose its value to be one. An illustration of this parameter setting procedure is shown in the electronic supplementary material.

## 2.7. Detecting outliers with Gaussian density estimation

As a benchmark method for outlier detection, we will fit a single Gaussian density to the data. We choose a single Gaussian density other than a mixture of Gaussian densities to detect outliers, as the latter is known to overfit a cluster of closely knit outliers, which will make them hard to detect [10]. The single Gaussian density estimation method models the dataset to be normally distributed about its means in the form of a multivariate Gaussian distribution. The probability distribution function is described as follows:

$$f(\mathbf{x}) = \frac{1}{(2\pi)^{m/2}|\Sigma|} \exp\left[-\frac{1}{2}(\mathbf{x} - \boldsymbol{\mu})\Sigma^{-1}(\mathbf{x} - \boldsymbol{\mu})\right], \tag{2.4}$$

where $|\Sigma|$ denotes the determinant of the covariance matrix, $\boldsymbol{\mu}$ is the $m$-dimensional sample mean vector and $\Sigma$ is the $m \times m$ sample covariance matrix. The term in the exponential is half the squared Mahalanobis distance of the sample $\mathbf{x}$ to the mean $\boldsymbol{\mu}$. The Mahalanobis distance can be used as the outlier score of each observation $\mathbf{x}$ and it is computed as follows:

$$\mathrm{MD}(\mathbf{x}, \boldsymbol{\mu}, \Sigma) = \sqrt{(\mathbf{x} - \boldsymbol{\mu})^T \Sigma^{-1}(\mathbf{x} - \boldsymbol{\mu})}. \tag{2.5}$$

Then the Gaussian density estimation method for outlier detection consists of finding the Mahalanobis distance to the sample mean for each observation in the dataset. Then the points that have the highest distance will have the lowest likelihood $f(\mathbf{x})$ and these samples are considered to be outliers. The Mahalanobis distance to the mean has shown promising results in intrusion and outlier detection by [21,22].

## 2.8. Traditional outlier detection methods

Methods such as median absolute deviation (MAD) and boxplot (BP) have been applied successfully to gene expression datasets to detect outliers [4]. Such methods are non-parametric and can detect outliers in the absence of any assumptions about the distribution of the data. BP method needs to find the lower quartile (25th percentile) and the upper quartile (75th percentile) of a specific sample which consists of a collection of gene expressions. Outlier genes of a specific sample are the data points that are above the upper fence or the data points below the lower fence. The upper fence is 1.5 times the interquartile range (IQR) above the upper quartile and the lower fence is 1.5 the IQR lower than the lower quartile. The IQR is defined by the difference between the upper and lower quartile. The BP method assigns a sample as an outlier when the number of outlier genes for that sample are greater than a pre-defined

threshold. The MAD outlier detection method is implemented by first finding the median of all genes of a sample then the MAD of all the genes from the median is calculated by:

$$\mathrm{MAD}_i = \mathrm{median}\big(|M_i - \mathrm{median}(M_i)|\big)$$

this is found for each sample $i$. The MAD can be thought of an outlier score for each sample. Therefore, samples with MAD higher than a pre-defined threshold are assigned to be outliers.

## 2.9. Datasets and data preparation

We demonstrate our results on three gene expression datasets, which are all publicly available. They are introduced as follows:

(i) Colon cancer dataset from [23]. It consists of a normalized dataset that contains 62 samples with gene expression levels of the 2000 genes with highest minimal intensity across the samples. The 62 samples are comprised of 40 tumour samples and 22 normal samples. A total of 40 patients are considered in their study and each tumour sample is taken from a different patient. In this study, we will only take into consideration the 40 tumour samples. The author of the data has shown that tumour samples of patients number: 2, 30, 33, 36 and 37 are outliers. They proved this by finding a muscle index for each of the 40 tumour samples and the 22 normal samples, taken from normal non-cancerous tissue. By taking into consideration that colon cancer samples mostly contain epithelial cells, which contain no muscle tissue, a high muscle index suggests a tissue being highly heterogeneous thus being a misleading tumour sample. Samples of patients: 2, 30, 33, 36 and 37 have muscle index that lies in the range of the muscle index of normal samples, thus being considered as outliers. In our work, we want to retain the genes that contain most of the information. Thus, we pre-process the data by retaining only the 700 most variable genes across samples. The number of most variable genes to keep is chosen to retain more than 85% of the total variance (sum of the variance of each gene in the dataset). The data are quantile normalized in the same way as [24], to reduce the skew of the microarray data to high expression levels, as recommended by [25].

(ii) TCGA breast cancer dataset, gathered from UCSC Xena browser [26]. Dataset consists of gene expression at transcription level, expressed as $\log_2(x + 1)$ transformed RSEM normalized RNA-sequencing counts. The TCGA dataset contains 20 530 genes with 1218 samples, 600 of which are patients with estrogen receptor (ER) positive status and 179 with ER-negative status, the remaining 439 samples do not have labels for ER thus are discarded. We sample 100 ER-positive samples from the 600 and five ER-negative samples from the 179. We repeat the random sampling process 30 times to have 30 datasets that will be used for outlier detection. For each of the datasets, the five ER-negative samples are considered to be the outliers. The choice of 100 ER+, five ER− and 30 random sampling repetitions do not affect the conclusions we get from our results. Conclusions will be consistent as long as the datasets are constructed with the reasoning that outlier samples need to be a small fraction of the overall dataset, and that we need to repeat the random sampling process so that we test the used methods on datasets with the same structure but with potentially different samples.

Since the number of genes is considerably large (greater than 20 000), it is necessary to diminish the dimensionality of the data to reduce the time of computation of the robust subspace methods and to get more stable results. Therefore, the data are filtered to retain the most variable genes, this is a commonly used pre-processing procedure for machine learning algorithms applied to genomic datasets [27] to choose the most informative genes. The number of genes to retain is chosen to trade-off between the time of computation and the fraction of the total variance explained by the chosen genes. For each of the 30 datasets, the 2000 most variable genes are retained. Note that choosing a number of genes greater than 2000 does not change the conclusion deduced from the results; it only increases the time needed for computation.

(iii) Single-cell dataset consisting of single-cell measurements of mouse embryonic stem cells at three different stages of the cell cycle (G1,S,G2M) gathered from [28]. The dataset consists of log-transformed normalized count values of gene expressions measured by single-cell RNA-seq for 8989 genes. There are a total of 182 cells, of which 59 are in G1, 58 in S and 65 in G2M. We build a dataset that consists of both the 59 G1 cells and 6 randomly sampled cells from the 65 G2M population. We repeat this random sampling process to gather 30 datasets that have six

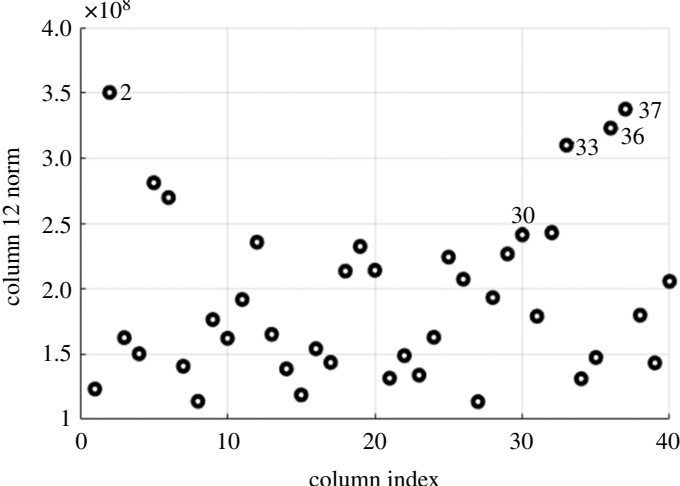

**Figure 1.** Inspecting $l_2$ norm of columns of $\hat{C}$. The labelled samples are the outlier samples found by the authors of the data in [23]. The figure shows that patients (2,33,36,37) are detected as outlier, except patient 30. This method differs from the $\hat{L}$ method for outlier detection in that we need to choose the threshold by having *a priori* knowledge of the fraction of outliers.

different G2M cells in each instance. For each of the datasets, the six G2M cells are taken to be the outliers. For each of the 30 constructed datasets, the 1000 most variable genes are retained following the same reasoning explained for the breast cancer dataset.

# 3. Results

## 3.1. Outlier detection on colon cancer dataset

After finding a suitable $\lambda$ with the procedure shown in electronic supplementary material, outliers are detected by inspecting the $l_2$ norms of the columns of $\hat{C}$. We expect that the columns of $\hat{C}$ that correspond to outliers to have higher $l_2$ norm than the non-outlier samples. From figure 1, we can see that the four highest points are actually four out of the five known outliers. Using the $\hat{L}$ method for outlier detection, we detect nine outliers in the minority cluster, four of which are the same true outlier samples detected by the $\hat{C}$ method and five are false positives.

We apply GOP to the colon cancer dataset to detect outlier samples. Outliers are detected using the $\hat{L}$ method as explained in §2.5. The regularization parameters $\lambda$ and $\alpha$ are tuned as explained in §2.6. Using $\alpha$ as 1 and optimal $\lambda$, four outliers are detected and they are part of the five known outliers from the author of the data [23]. This gives better outlier detection than OP which finds the same four outliers but has five false positives. GOP performs the same as OP when the $\hat{C}$ matrix is used, but we did not have to choose a suitable fraction of outliers. It should be noted that the same four out of the five known outliers are picked up by average hierarchical clustering used in [4]. To further compare the outlier detection capability of OP and GOP we project $\hat{L}$ (recovered from each method using optimal regularization parameters) on its first two principal directions, and we find the Mahalanobis distance (MD) between the two cluster centres found by $k$-means on the projection. Furthermore, the capability of capturing the outliers visually is compared for GOP, OP, PCA and t-distributed stochastic neighbour embedding (t-SNE) by finding their two-dimensional embedding. We get an MD of 2.806 for $\hat{L}$ from GOP and 1.8973 for $\hat{L}$ from OP and an MD of 1.7839 and 1.6777 for PCA and t-SNE, respectively. We can see the greater separation between outlier and main samples from GOP in figure 2. In conclusion, $\hat{L}$ recovered by GOP gives better separation between main samples and outlier samples and this gives fewer false positives when detecting outliers. Although, GOP gives less false positives than OP it still missed the same outlier that OP missed.

## 3.2. Outlier detection capability on breast cancer dataset

Given the 30 sampled datasets with 105 samples, we need to detect the five ER-negative as outliers using the five methods: OP, GOP, Gaussian density, MAD and BP. Each of the 30 datasets will be supplied as input for the aforementioned methods. We detect outliers for OP and GOP using the $l_2$ norms of the $\hat{C}$

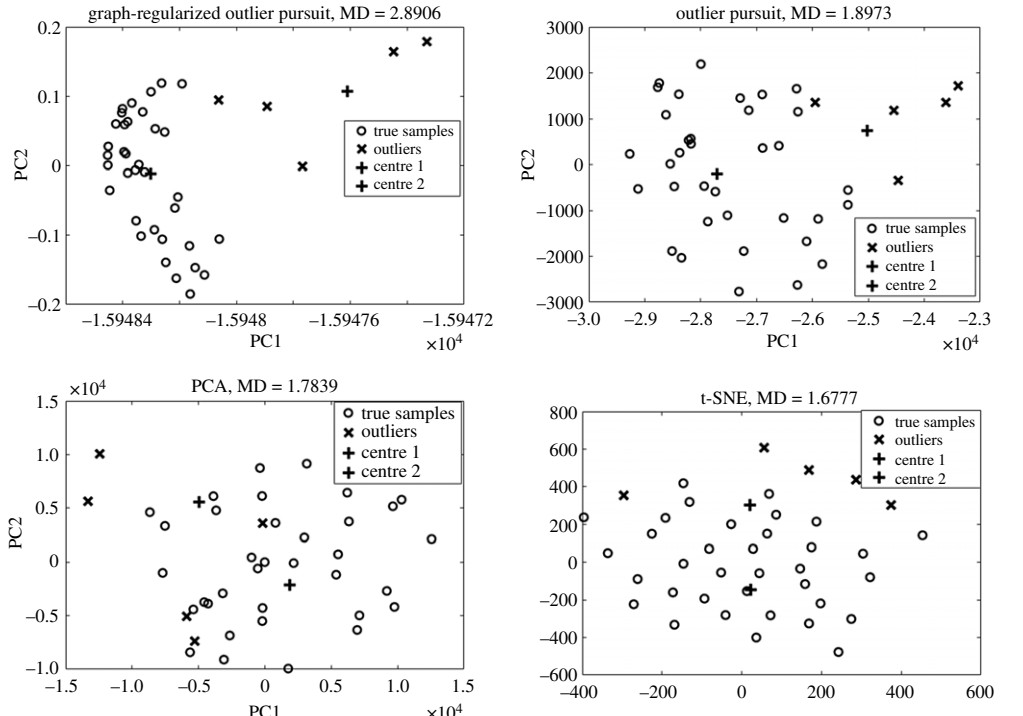

**Figure 2.** Two-dimensional visualization found by GOP, OP, PCA and t-SNE. For GOP and OP, we project $\hat{L}$ onto its first two principal directions. PCA and t-SNE are applied directly to the colon cancer dataset. Figure shows that the separation between main and outlier samples is greater in the subspace found by GOP.

matrix and will record the number of false positives encountered before finding all the five outlier samples. For the Gaussian density method, we will use the Mahalanobis distance to the sample mean as an outlier score for each sample and record the number of false positives needed to recover all the five outliers. For the MAD method, the outlier score will be the MAD of each sample and record the number of false positives to detect all the five known outliers. For the BP method, we will use the number of outlier genes as an outliers score for each sample and we will record the false positives encountered to detect all the five known outliers. False positives from the five methods will be recorded for all the 30 randomly sampled datasets. This experiment will be repeated by changing the dimensionality of the 30 datasets. The dimensionality will be changed by using the most variable genes across samples. The false positives for the 30 datasets will be recorded for GOP, OP and the Gaussian density method at 25, 50, 80, 95, 100 and 200 dimensions. At 200 dimensions, MAD and BP are added to validate the performance of GOP and OP. The results are shown in figure 3. We can see that for 25 dimensions the number of false positives is high for all three methods, and it decreases when the number of dimensions increases to 50 and 80. This is due to the fact that there is more useful information injected by the added dimensions. At 95, 100 and 200 dimensions, the performance of the Gaussian density method is degraded as there are not enough samples compared to the number of dimensions. However, we can see that for both OP and GOP, the outlier detection keeps improving by increasing the dimensions of the input dataset. Furthermore, we note that GOP has a smaller median of false positives encountered to detect all five outliers on each chosen dimension when compared to OP. Finally, we can see at 200 dimensions that MAD and BP record a much higher median percentage of false positives compared to GOP and OP.

### 3.2.1. Low-dimensional embedding outlier detection and visualization

We showed in the previous subsection that the graph regularizer can enhance the outlier detection performance using the $\hat{C}$ matrix. In this subsection, we show that the same can be achieved for the separation of outlier and main samples in the recovered low-rank matrix $\hat{L}$ and how this better separation can be visualized in two dimensions. We recover $\hat{L}$ for each of the 30 randomly sampled datasets. Next, we perform $k$-means clustering with two clusters on the projection of $\hat{L}$ and find the F-score. This is performed on all the low-dimensional embeddings of GOP, OP, PCA and t-SNE. We

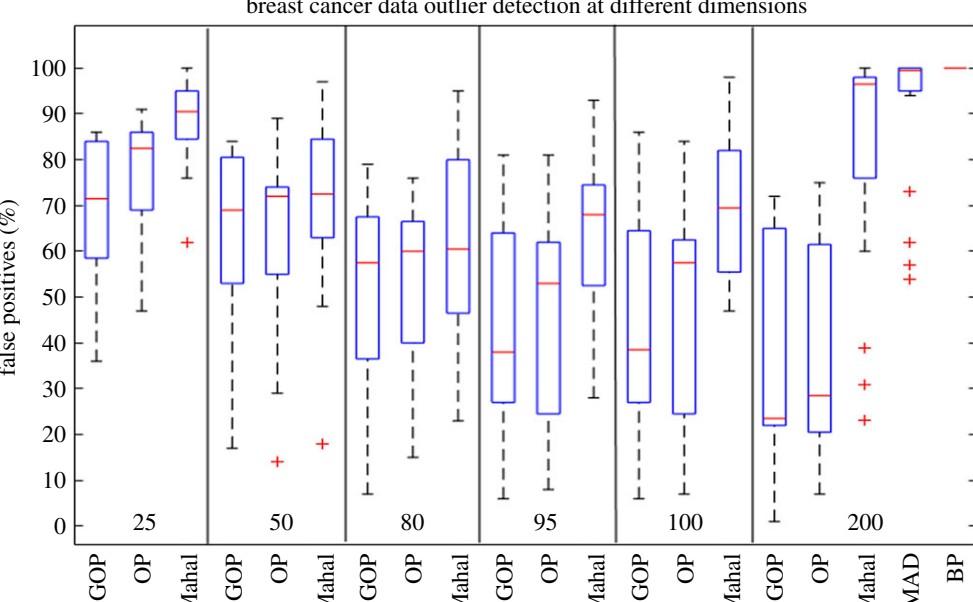

**Figure 3.** Boxplots comparing the number of false positives encountered to detect all five outliers in the 30 instances of the breast cancer dataset. Each of the six subdivisions of the figure represents running GOP, OP and the Gaussian density method for all 30 datasets at a specific dimension. The dimension used is indicated at the bottom of each subdivision. The horizontal line in each boxplot corresponds to the median of false positives. We find that the Gaussian density method finds on average more false positives than both OP and GOP. Moreover, we can see that the Gaussian density method suffers from the curse of dimensionality whereas the subspace methods are robust to high-dimensional datasets. Furthermore, we note that GOP detects less false positives on average than both methods, showing that the outlier detection has benefited from the graph regularization.

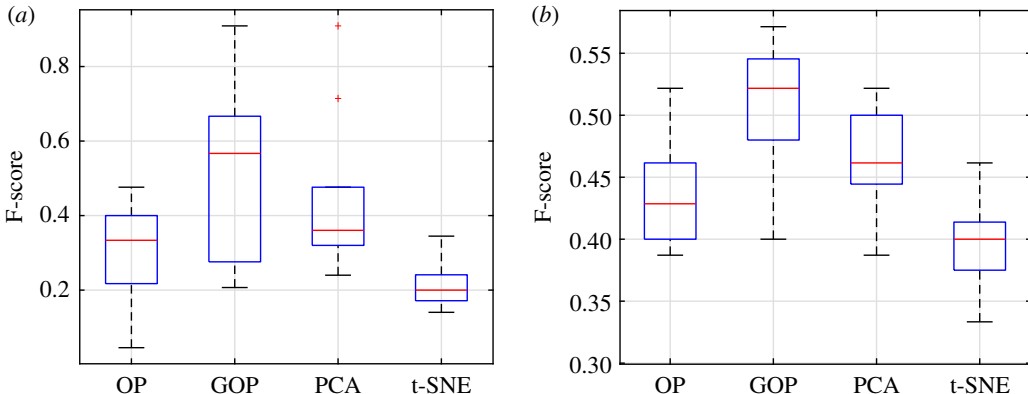

**Figure 4.** (*a*) (Breast cancer dataset) F-score of *k*-means clustering for all dimensionality reduction methods, found on the 30 instances of the breast cancer dataset. Each boxplot shows the F-score for all 30 randomly sampled datasets by the corresponding dimensionality reduction method. We can see that GOP has a considerably higher median F-score compared to all other methods. (*b*) (Single-cell dataset) F-score of *k*-means for all dimensionality reduction methods applied to the 30 instances of the single-cell dataset. We can see that GOP gives the best F-score in its low-dimensional embedding compared to all other methods.

supply the same input to all the dimensionality reduction methods which is the 105 samples after filtering its genes to the 2000 most variable genes across samples. From figure 4*a*, it is observed that the F-score found on the GOP low-dimensional embedding is greater than all other methods. In this case, the F-score of GOP is highest because it detects more true positives and less false positives than all other methods. This greater capability to detect outliers in the low-dimensional embedding can be further seen visually by projection of $\hat{L}$ onto its first two principal directions. The visualization in two dimensions is shown in figure 5. We can observe that GOP gives better separation of the five ER− samples and the 100 ER+ samples in its two-dimensional projection. In the t-SNE two-dimensional embedding, the separation is also seen clearly. However, we note from the F-score that

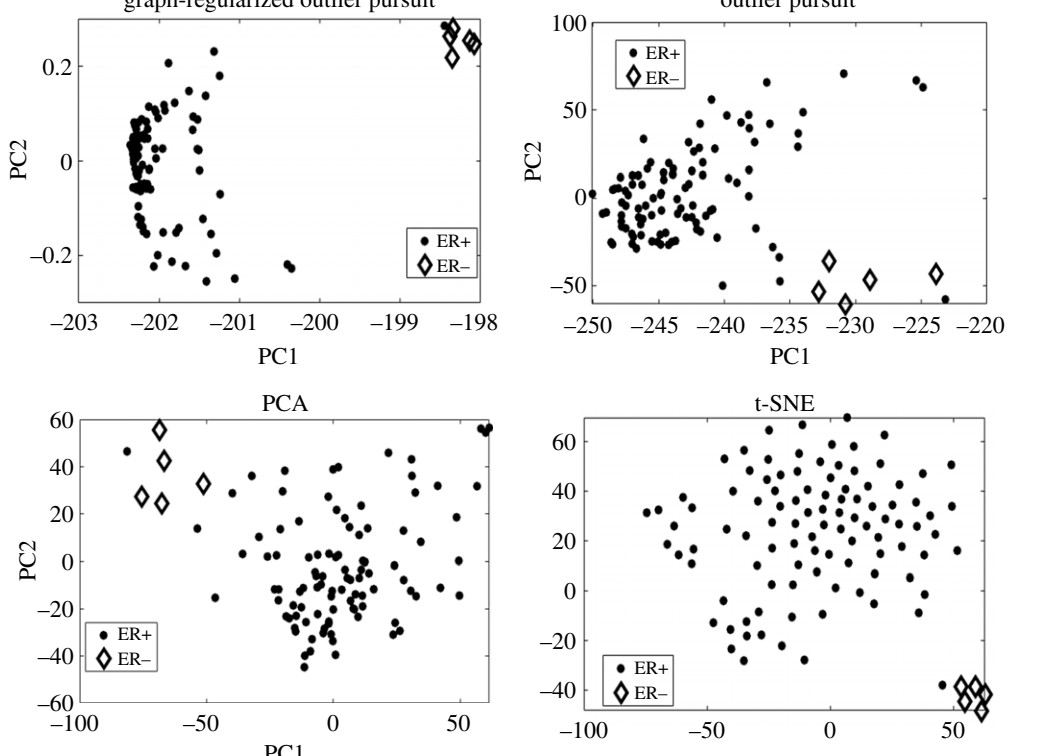

**Figure 5.** Visualization of two-dimensional embedding for each dimensionality reduction method on a chosen instance of the breast cancer dataset. The figure shows the enhanced separation of main and outlier samples in the GOP embedding compared to OP, PCA and t-SNE.

two cluster $k$-means on this space fail to find the outliers and main samples accurately. In the case of GOP, we can visually observe the separation and quantitatively measure this separation using a standard clustering technique such as $k$-means.

## 3.3. Outlier detection capability on single-cell dataset

In this section, we compare the outlier detection performance of GOP, OP, Gaussian density, MAD and BP methods on the 30 randomly sampled single-cell datasets constructed as discussed in §2.9. The task consists of finding the six G2M cells in the population of 59 G1 cells as outliers in each of the 30 datasets. Similar to the breast cancer dataset, we record the number of false positives to detect all the known six outliers using the aforementioned methods. This is repeated for six different dimensions by retaining the most variable genes across samples. The different dimensions are: 2, 20, 30, 50, 60 and 70. Figure 6 shows the outlier detection results for the single-cell dataset. We can see that the outlier detection performance of the Gaussian density method improves when increasing the dimensionality from 2 to 20. However, the number of false positives starts to increase monotonically by increasing the dimensions from 20 to 70. Moreover, we note that the performance of the robust subspace methods improves with the increase in dimensionality, showing that they are effective in filtering out the noise and extracting useful information from high-dimensional datasets. They avoid falling into the curse of dimensionality because the data matrix is modelled to be a low-rank matrix that is corrupted by a column-sparse matrix modelling the outliers. Thus, the robust subspace methods work in a reduced dimensional space which help it to circumvent the curse of dimensionality. Furthermore, we note that GOP outperforms OP on all dimensions, showing that the graph regularizer is also beneficial on the single-cell dataset. Finally, we can see that at 70 dimensions MAD and BP record a considerably higher percentage of false positives when compared with GOP and OP.

### 3.3.1. Low-dimensional embedding outlier detection and visualization

Here, we show the F-score of $k$-means with two cluster centres on the low-dimensional projections of GOP, OP, PCA and t-SNE. We give as input to each dimensionality reduction technique, an instance of

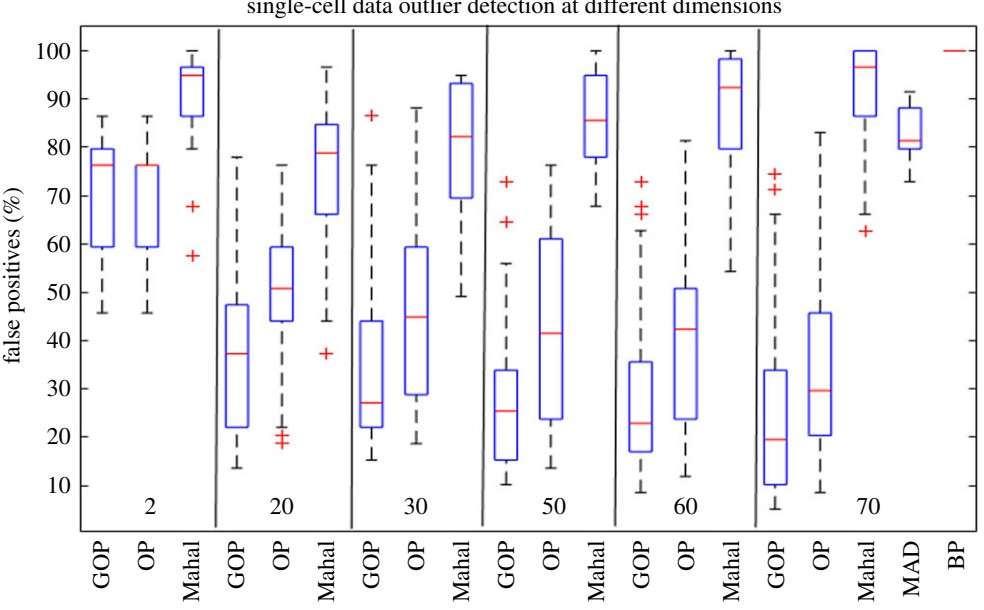

**Figure 6.** Boxplots comparing the number of false positives encountered to detect all six outliers in the 30 instances of the single-cell dataset. We inspect the number of false positives at six different dimensions. We note that the performance of the Gaussian density method improves by increasing the dimensions from 2 to 20, as this adds more useful information to the dataset. However, it starts to degrade when increasing further. Moreover, we can see that the outlier detection performance of the robust subspace methods improves with the increase in dimensionality. Furthermore, we can see that GOP detects less median of false positives than OP at every dimension chosen.

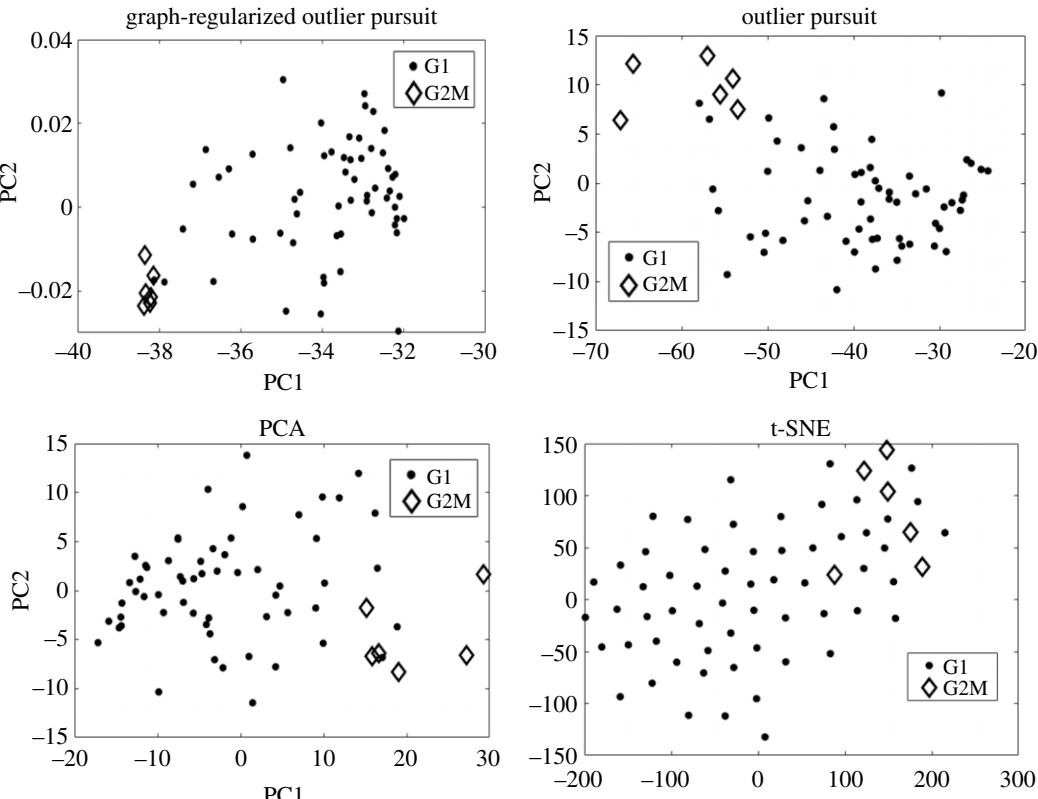

**Figure 7.** Two-dimensional visualization of the dimensionality reduction methods for a specific instance of the single-cell dataset. Figure shows the enhanced visualization property of GOP compared to OP, PCA and t-SNE.

the single-cell data after filtering it to its 1000 most variable genes. We find the F-score for all the 30 different instances of the single-cell data. As seen from figure 4*b*, the greater F-score of GOP indicates that the outlier samples are better separated from the main samples in the lower-dimensional embedding of GOP. In this case, the F-score of GOP is highest because it detects less false positives than all other methods. All the methods generally detect all the known six outliers. From figure 7, we can see that GOP separates the outlier and main samples better than the other dimensionality reduction methods. This gives GOP an enhanced visualization property compared to other methods.

# 4. Discussion

The graph-regularized method introduced in this paper to detect outlying samples has more features compared to similar work found in the literature. With GOP we can detect outliers in two distinct ways: outlier ranking through the $\hat{C}$ matrix and clustering through the $\hat{L}$ matrix. Moreover, GOP is also a dimensionality reduction technique which makes it visualizable in a two-dimensional space. Other methods such as [29,30] only devised an outlier ranking procedure by taking measures of global similarity between samples. This makes their method only capture outliers, but makes it harder to identify different subgroups. The identification of subgroups is leveraged by clustering techniques more than outlier ranking techniques. There are previous papers that used clustering techniques to identify similarity of samples in gene expression data, which are reviewed in [31]. However, they do not give the capability to visualize the data and do not give an outlyingness ranking of the samples. To the best of our knowledge GOP in the only method that combines both outlier ranking of samples with clustering and visualization.

# 5. Conclusion

In this paper, we develop an outlier detection framework for functional genomics data using structured low-rank matrix approximation methods. We derive an optimization algorithm to estimate the decomposition of the data matrix into a low rank and column-sparse matrices and explore two ways of extracting outliers from the decomposition. The formulation also includes a regularizer based on the graph Laplacian of the data. Using transcriptomic data from bulk and single-cell measurements, we show that the method reliably detects injected outliers, particularly when the graph regularizer is used. Most importantly, when compared to a density-based method of thresholding the Mahalanobis distance, the method we advance does not fail with increasing dimensions. Thus finding the low-rank subspace, in this case, it has shown to circumvent the curse of dimensionality. The graph regularizer used in this study is based on affinity (or neighbourhood) of the data. However, this can be a convenient handle to inject prior knowledge into the problem domain. Thus our current work focuses on the use of archived prior knowledge (interaction networks of the resulting proteins, for example) as regularizers.

Data accessibility. All data used are publicly available. Matlab code is available on GitHub. https://github.com/omarshetta/Manuscript_Royal_Society.

Authors' contributions. O.S. and M.N. jointly designed the study, O.S. carried out the complete simulations and both authors interpreted the results and wrote the manuscript.

Competing interests. We declare we have no competing interests.

Funding. O.S. was supported by Engineering and Physical Sciences Research Council (EPSRC) and M.N.'s contribution was funded by the EPSRC project: from data to inference (EP/N014189/1).

Acknowledgements. No one contributed to the study that does not meet authorship criteria.

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
