## [Reviewer comments · Royal Society Open Science]

Review History

RSOS-190714.R0 (Original submission)

Review form: Reviewer 1

Is the manuscript scientifically sound in its present form?

Yes

Are the interpretations and conclusions justified by the results?

No

Is the language acceptable?

No

Do you have any ethical concerns with this paper?

No

Have you any concerns about statistical analyses in this paper?

No

Recommendation?

Major revision is needed (please make suggestions in comments)

Comments to the Author(s)

This manuscript introduces 2 new methods for introducing data outlier points, e.g. in gene expression data sets.

Curating noisy and possibly erroneous experimental data sets is an important task that is often not properly mentioned in methods sections of published paper that present analyses of transcriptomic data sets.

When outlier data points are not removed, the downstream analysis may be heavily affected.

Possibly, the new methods presented in this manuscript may be beneficial in this respect.

However, it is unclear to me how superior the methods are compared to "traditional" methods used to detect outliers such as MAD or boxplot analysis of gene expression values, see my point (7).

Also see my first point below about the usefulness of the current implementation.

I was surprised that this manuscript submitted from a respected university in UK contains many incomplete sentences and incorrect grammar (verb tenses, use of singular/plural form).

The supporting material of the manuscript seems to be in better shape than the main manuscript.

(1) (a) I had a look at the code deposited at Github.

I didn't have the impression that the authors would like others to use their code.

(a) The code is almost free from any comments which tell the user what is done.

(b) There are no instructions in which sequence the scripts should be run, or if there exists a "main" script that executes the others.

I suggest that there should be README files explaining users how to execute these runs so that they can at least reproduce the results of this manuscript.

(b) It would be most convenient for users in the biological field if the authors also provided a version in R language and would submit it to the Bioconductor suite as an R package. This platform has a quasi monopoly in this field.

Of course, the authors may be unwilling to do so, but then their methods will likely not become popular.

(2) p.2 line 20 "This work shows that ... protein concentrations can be predicted from mRNA levels."

This is not true in a general sense, see e.g. Fig. 2(c) in

<https://stke.sciencemag.org/content/3/104/ra3.full>

That figure shows that there are proteins where protein regulation follows mRNA regulation, but there are also many proteins where either protein or mRNA levels are unchanged during the cell cycle while the other one is regulated.

(3) p.3 line 44: How is $|M|_{\subscript{F}}$ defined? This seems to be math jargon that is not understandable to the general audience working with gene expression data.

Below algorithm 1, it should be added that $L_{\superscript{k}}$ stands for L at iteration k.

(4) p.4 line 25 "by first finding the nearest neighbors of each sample" is unclear.

How many nearest neighbors do you consider?

Or do you consider all points and weight them by W_{ij} ?

(5) p.7 line 26 vs. line 35: Do you consider 2000 or 700 genes?

line 35: I find it strange that quantile normalization is applied before removing outliers.

(6) p.7 line 55: Sentence "After finding ..." is incomplete.

line 57: there is no Fig. 1(c).

(7) MAJOR POINT: the authors considered a TCGA dataset with 100 ER positive samples and 5 ER negative samples.

(a) Fig. 3 shows that - in the best case with 200 dimensions - about 30 ER positive samples are identified as "false positives" before all 5 ER negative samples were detected. I don't find this performance very convincing.

I wonder whether "traditional" methods used to detect outliers such as MAD or boxplot analysis of gene expression values would give better results?

(b) The y-axis of Fig. 3 should either be labeled "absolute number of false positives" or "false positives (%)". I believe in the present case it doesn't matter, but the current label forces the reader to go back to the methods section and check how the TCGA dataset was constructed.

(8) The author list of ref. (4) is incomplete.

Minor points

(5) p.2 line 17: techniques to outliers -> techniques to determine outliers

(6) p.2 line 31: number of feature increase -> number of features increases
line 31/32: reword "the space the data lives in". To my understanding, data is not a living object.

(7) p.2 line 51: "gives it much better computational efficiency from the state of the art".
is unclear. Reword.
line 52: the sentence "this model same as standard PCA" is incomplete.

(8) p.3 line 28/29. Sentence "Where the identify ... of the data matrix" is incomplete.

(9) p.4 last line: a citation to the ADAM method should be added.

(10) p.6 line 21: sentence "Which suffers from ... of the domain." is incomplete
line 22: will decided -> will decide
line 23: corresponds -> correspond
line 23: Sentence "Therefore, giving a cut-off ..." is incomplete.

(11) p.8 line 26: by the a author -> by the authors
line 47: from these figure -> from these figures

(12) p.13 line 45: Sentence "Compared to [26] and [27] ... between samples" is incomplete.

Review form: Reviewer 2 (Ahmad Barghash)

Is the manuscript scientifically sound in its present form?

Yes

Are the interpretations and conclusions justified by the results?

No

Is the language acceptable?

Yes

Do you have any ethical concerns with this paper?

No

Have you any concerns about statistical analyses in this paper?

Yes

Recommendation?

Accept with minor revision (please list in comments)

Comments to the Author(s)

0. The authors should mention clearly that outlier genes can not be detected using this algorithm as the references they chose frequently discussed outlier samples and genes

1. Tuning the value of lambda does not appear to be a straight forward approach. The authors mentioned that the value of lambda "needs to be chosen in such a way that the number of outliers are not too large". I believe "too large" is too general and can be tumor specific, user specific, case specific, .. etc. Additionally, relying on the long mathematical approach to chose lambda each time a user wants to use the presented algorithm would drive the possible users from biology or pharmacy away. the authors should suggest a simpler way to chose the optimal lambda

2. it is not clear why the authors used the known outliers in tumor samples in the colon dataset while reference [4] in the manuscript presented the history of the chosen colon dataset and 9 outliers are widely known not only 5. Have the authors tested their presented algorithm on the outliers in normal samples too?

3. choosing the "the most variables genes across sample" is unclear. Why the authors had to discard a huge fraction of the genes? how did the authors decide about how many "variables genes across sample" are needed? For example, in one dataset they chose the top 700 and in another the top 2000.

4. The authors should mention clearly why they chose to sample TCGA datasets by 100+ and 5-samples and also why exactly 30 datasets. was it s trial-error approach to chose the 100, 5, and 30?

5. The TCGA dataset has another 429 samples that the authors did not mention. datasets should be described in details even if parts of them will be discarded later

6. The single cell measurements can be for gene expression or DNA methylation for example. the users should mention in the datasets section all details about the third dataset.

7. the authors should mention if they used RAW or normalized datasets in their testing

8. In the results and discussion sections, the authors should stress on the finding that GOP will have less false positives but still will probably miss any outlier that OP might miss.

9. the authors should mention that basic clustering methods have had similar performance to their presented algorithm. For example, in [4] it is presented that average hierarchical clustering missed only one outlier sample when applied to the same colon dataset the authors used

10. in Figure 3, what are the 6 boxes referring to? 6 runs for the 30 datasets?

11. The authors mentioned that "the performance of the robust subspace methods improves with the increase in dimensionality," how can it avoid falling in the curse of dimensionality?

12. Consistency issues:

A. OP and GOP results always appeared in the same section except in the analysis of the colon dataset they were separated in two sections.

B. PCA and t-SNE were tested on the TCGA dataset only

3. in the datasets section, the number of chosen "most variables genes" was mentioned in the colon dataset section but for other datasets it was mentioned in the results. It should be in the same place for all of the datasets

13. In the references, reference [4] is missing one author name. I suggest that the authors double check the whole reference list for missing authors in other references. Additionally,

Decision letter (RSOS-190714.R0)

23-Sep-2019

Dear Mr Shetta,

The editors assigned to your paper ("Robust Subspace Methods for Outlier Detection in Genomic Data Circumvents the Curse of Dimensionality") have now received comments from reviewers. We would like you to revise your paper in accordance with the referee and Associate Editor suggestions which can be found below (not including confidential reports to the Editor). Please note this decision does not guarantee eventual acceptance.

Please submit a copy of your revised paper before 16-Oct-2019. Please note that the revision deadline will expire at 00.00am on this date. If we do not hear from you within this time then it will be assumed that the paper has been withdrawn. In exceptional circumstances, extensions may be possible if agreed with the Editorial Office in advance. We do not allow multiple rounds of revision so we urge you to make every effort to fully address all of the comments at this stage. If deemed necessary by the Editors, your manuscript will be sent back to one or more of the original reviewers for assessment. If the original reviewers are not available, we may invite new reviewers.

- Data accessibility

It is a condition of publication that all supporting data are made available either as supplementary information or preferably in a suitable permanent repository. The data

accessibility section should state where the article's supporting data can be accessed. This section should also include details, where possible of where to access other relevant research materials such as statistical tools, protocols, software etc can be accessed. If the data have been deposited in an external repository this section should list the database, accession number and link to the DOI for all data from the article that have been made publicly available. Data sets that have been deposited in an external repository and have a DOI should also be appropriately cited in the manuscript and included in the reference list.

If you wish to submit your supporting data or code to Dryad (<http://datadryad.org/>), or modify your current submission to dryad, please use the following link:
<http://datadryad.org/submit?journalID=RSOS&manu=RSOS-190714>

- **Competing interests**

- **Authors' contributions**

- **Acknowledgements**

- **Funding statement**

on behalf of Professor Andrew Teschendorff (Associate Editor) and Marta Kwiatkowska (Subject Editor)
openscience@royalsociety.org

Associate Editor's comments (Professor Andrew Teschendorff):

Associate Editor: 1

Comments to the Author:

Please submit a revised version addressing the reviewer's comments

Comments to Author:

Reviewers' Comments to Author:

Reviewer: 1

Comments to the Author(s)

This manuscript introduces 2 new methods for introducing data outlier points, e.g. in gene expression data sets.

Curating noisy and possibly erroneous experimental data sets is an important task that is often not properly mentioned in methods sections of published paper that present analyses of transcriptomic data sets.

When outlier data points are not removed, the downstream analysis may be heavily affected.

Possibly, the new methods presented in this manuscript may be beneficial in this respect.

However, it is unclear to me how superior the methods are compared to "traditional" methods used to detect outliers such as MAD or boxplot analysis of gene expression values, see my point (7).

Also see my first point below about the usefulness of the current implementation.

I was surprised that this manuscript submitted from a respected university in UK contains many incomplete sentences and incorrect grammar (verb tenses, use of singular/plural form).

The supporting material of the manuscript seems to be in better shape than the main manuscript.

(1) (a) I had a look at the code deposited at Github.

I didn't have the impression that the authors would like others to use their code.

(a) The code is almost free from any comments which tell the user what is done.

(b) There are no instructions in which sequence the scripts should be run, or if there exists a "main" script that executes the others.

I suggest that there should be README files explaining users how to execute these runs so that they can at least reproduce the results of this manuscript.

(b) It would be most convenient for users in the biological field if the authors also provided a version in R language and would submit it to the Bioconductor suite as an R package. This platform has a quasi monopoly in this field.

Of course, the authors may be unwilling to do so, but then their methods will likely not become popular.

(2) p.2 line 20 "This work shows that ... protein concentrations can be predicted from mRNA levels."

This is not true in a general sense, see e.g. Fig. 2(c) in

<https://stke.sciencemag.org/content/3/104/ra3.full>

That figure shows that there are proteins where protein regulation follows mRNA regulation, but there are also many proteins where either protein or mRNA levels are unchanged during the cell cycle while the other one is regulated.

(3) p.3 line 44: How is $|M|_{\subscript{F}}$ defined? This seems to be math jargon that is not understandable to the general audience working with gene expression data.

Below algorithm 1, it should be added that $L_{\superscript{k}}$ stands for L at iteration k.

(4) p.4 line 25 "by first finding the nearest neighbors of each sample" is unclear.

How many nearest neighbors do you consider?
Or do you consider all points and weight them by W_{ij} ?

(5) p.7 line 26 vs. line 35: Do you consider 2000 or 700 genes?

line 35: I find it strange that quantile normalization is applied before removing outliers.

(6) p.7 line 55: Sentence "After finding ..." is incomplete.
line 57: there is no Fig. 1(c).

(7) MAJOR POINT: the authors considered a TCGA dataset with 100 ER positive samples and 5 ER negative samples.

(a) Fig. 3 shows that - in the best case with 200 dimensions - about 30 ER positive samples are identified as "false positives" before all 5 ER negative samples were detected. I don't find this performance very convincing.

I wonder whether "traditional" methods used to detect outliers such as MAD or boxplot analysis of gene expression values would give better results?

(b) The y-axis of Fig. 3 should either be labeled "absolute number of false positives" or "false positives (%)". I believe in the present case it doesn't matter, but the current label forces the reader to go back to the methods section and check how the TCGA dataset was constructed.

(8) The author list of ref. (4) is incomplete.

Minor points

(5) p.2 line 17: techniques to outliers -> techniques to determine outliers

(6) p.2 line 31: number of feature increase -> number of features increases
line 31/32: reword "the space the data lives in". To my understanding, data is not a living object.

(7) p.2 line 51: "gives it much better computational efficiency from the state of the art" is unclear. Reword.
line 52: the sentence "this model same as standard PCA" is incomplete.

(8) p.3 line 28/29. Sentence "Where the identify ... of the data matrix" is incomplete.

(9) p.4 last line: a citation to the ADAM method should be added.

(10) p.6 line 21: sentence "Which suffers from ... of the domain." is incomplete
line 22: will decided -> will decide
line 23: corresponds -> correspond
line 23: Sentence "Therefore, giving a cut-off ..." is incomplete.

(11) p.8 line 26: by the a author -> by the authors
line 47: from these figure -> from these figures

(12) p.13 line 45: Sentence "Compared to [26] and [27] ... between samples" is incomplete.

Reviewer: 2

Comments to the Author(s)

0. The authors should mention clearly that outlier genes can not be detected using this algorithm as the references they chose frequently discussed outlier samples and genes

1. Tuning the value of lambda does not appear to be a straight forward approach. The authors mentioned that the value of lambda "needs to be chosen in such a way that the number of outliers are not too large". I believe "too large" is too general and can be tumor specific, user specific, case specific, .. etc. Additionally, relying on the long mathematical approach to chose lambda each time a user wants to use the presented algorithm would drive the possible users from biology or pharmacy away. the authors should suggest a simpler way to chose the optimal lambda
2. it is not clear why the authors used the known outliers in tumor samples in the colon dataset while reference [4] in the manuscript presented the history of the chosen colon dataset and 9 outliers are widely known not only 5. Have the authors tested their presented algorithm on the outliers in normal samples too?
3. choosing the "the most variables genes across sample" is unclear. Why the authors had to discard a huge fraction of the genes? how did the authors decide about how many "variables genes across sample" are needed? For example, in one dataset they chose the top 700 and in another the top 2000.
4. The authors should mention clearly why they chose to sample TCGA datasets by 100+ and 5-samples and also why exactly 30 datasets. was it s trial-error approach to chose the 100, 5, and 30?
5. The TCGA dataset has another 429 samples that the authors did not mention. datasets should be described in details even if parts of them will be discarded later
6. The single cell measurements can be for gene expression or DNA methylation for example. the users should mention in the datasets section all details about the third dataset.
7. the authors should mention if they used RAW or normalized datasets in their testing
8. In the results and discussion sections, the authors should stress on the finding that GOP will have less false positives but still will probably miss any outlier that OP might miss.
9. the authors should mention that basic clustering methods have had similar performance to their presented algorithm. For example, in [4] it is presented that average hierarchical clustering missed only one outlier sample when applied to the same colon dataset the authors used
10. in Figure 3, what are the 6 boxes referring to? 6 runs for the 30 datasets?
11. The authors mentioned that "the performance of the robust subspace methods improves with the increase in dimensionality, " how can it avoid falling in the curse of dimensionality?
12. Consistency issues:
 - A. OP and GOP results always appeared in the same section except in the analysis of the colon dataset they were separated in two sections.
 - B. PCA and t-SNE were tested on the TCGA dataset only
3. in the datasets section, the number of chosen "most variables genes" was mentioned in the colon dataset section but for other datasets it was mentioned in the results. It should be in the same place for all of the datasets
13. In the references, reference [4] is missing one author name. I suggest that the authors double check the whole reference list for missing authors in other references. Additionally,

Author's Response to Decision Letter for (RSOS-190714.R0)

See Appendix A.

RSOS-190714.R1 (Revision)

Review form: Reviewer 1

Is the manuscript scientifically sound in its present form?

Yes

Are the interpretations and conclusions justified by the results?

Yes

Is the language acceptable?

Yes

Do you have any ethical concerns with this paper?

No

Have you any concerns about statistical analyses in this paper?

No

Recommendation?

Accept as is

Comments to the Author(s)

The authors have appropriately addressed my points.

Review form: Reviewer 2 (Ahmad Barghash)

Is the manuscript scientifically sound in its present form?

Yes

Are the interpretations and conclusions justified by the results?

Yes

Is the language acceptable?

Yes

Do you have any ethical concerns with this paper?

No

Have you any concerns about statistical analyses in this paper?

Yes

Recommendation?

Accept as is

Comments to the Author(s)

The authors answered my previous comments and modified the manuscript accordingly. No further comments from my side

Decision letter (RSOS-190714.R1)

12-Dec-2019

Dear Mr Shetta,

It is a pleasure to accept your manuscript entitled "Robust Subspace Methods for Outlier Detection in Genomic Data Circumvents the Curse of Dimensionality" in its current form for publication in Royal Society Open Science. The comments of the reviewer(s) who reviewed your manuscript are included at the foot of this letter.

Kind regards,
Anita Kristiansen
Editorial Coordinator
Royal Society Open Science
openscience@royalsociety.org

on behalf of Professor Andrew Teschendorff (Associate Editor) and Marta Kwiatkowska (Subject Editor)
openscience@royalsociety.org

Subject Editor Comments to Author (Marta Kwiatkowska):
The referee comments have been appropriately addressed.

Reviewer comments to Author:

Reviewer: 1

Comments to the Author(s)

The authors have appropriately addressed my points.

Reviewer: 2

Comments to the Author(s)

The authors answered my previous comments and modified the manuscript accordingly. No further comments from my side

Appendix A

Response to Referees

We would like to thank the reviewers for their comments. Below are the responses to each comment of both reviewers. Overall we feel that the changes made have helped to dramatically improve the clarity of the paper. Line and page numbers referred to in the responses below are from the uploaded file ‘**Revised Manuscript Tracked Changes**’.

Response to Reviewer 1

Comment (1) (a) : *I had a look at the code deposited at Github. I didn't have the impression that the authors would like others to use their code. (a) The code is almost free from any comments which tell the user what is done. (b) There are no instructions in which sequence the scripts should be run, or if there exists a “main” script that executes the others. I suggest that there should be README files explaining users how to execute these runs so that they can at least reproduce the results of this manuscript.*

Response (1) : We agree that the code deposited was untidy and rushed (though correctly reproduces the quoted results). We have polished the newly supplied code.

- The code is now well commented.
- The folders of the three datasets are better organised.
- README files have been added to direct the user on how to replicate the results shown in the paper.

We thank the reviewer for their valuable feedback. We are currently working towards rewriting the code in a more popular language.

Comment (2): *p.2 line 20 “This work shows that ... protein concentrations can be predicted from mRNA levels.” This is not true in a general sense.*

Response (2) : Thank you for this clarification. In page 2 line 14-16 we have taken the reviewers comment into consideration and give a more specific description of what reference [6] has actually done. [6] has predicted protein concentrations in yeast cells by using mRNA abundance in addition with sequence derived features, using linear regression getting an r^2 of 0.86. Proteins that are post translationally regulated have high prediction error thus are labelled as outliers.

Comment (3) : *p.3 line 44: How is $\|M\|_F$ defined? This seems to be math jargon that is not understandable to the general audience working with gene expression data. Below algorithm 1, it should be added that L^k stands for L at iteration k .*

Responses (3) : We have added in page 4 lines 1 to 2 the definition for $\|M\|_F$ and that k in L^k stands for iteration index.

Comment (4) : *p.4 line 25 “by first finding the nearest neighbors of each sample” is unclear. How many nearest neighbors do you consider? Or do you consider all points and weight them by W_{ij} ?*

Response (4) : We have addressed this point in page 4 lines 15-20. Starting at “The graph which has nodes” to “outliers present in the data”.

Comment (5) : *p.7 line 26 vs. line 35: Do you consider 2000 or 700 genes? line 35: I find it strange that quantile normalization is applied before removing outliers.*

Response (5) : We have addressed this in section 2 (i) bullet point (i). We considered 700 most variable genes. We meant that 2000 genes were retained by the main author of the data, reference [22]. We used the 700 most variable genes in this gene set of 2000. This has been clarified in page 7 lines 32-37.

We quantile normalize our data following reference [32] where they used the same colon cancer dataset and applied a robust PCA method. This has been clarified in page 8 line 7, sentence starting “The data...” .

Comment (6) : *p.7 line 55: Sentence “After finding ...” is incomplete. line 57: there is no Fig. 1(c).*

Response (6) : Correction has been made in page 8 line 45 to 46. Figure label has been corrected in page 8 line 47.

Comment (7) : *MAJOR POINT: the authors considered a TCGA dataset with 100 ER positive samples and 5 ER negative samples. (a) Fig. 3 shows that - in the best case with 200 dimensions - about 30 ER positive samples are identified as “false positives” before all 5 ER negative samples were detected. I don’t find this performance very convincing. I wonder whether “traditional” methods used to detect outliers such as MAD or boxplot analysis of gene expression values would give better results?*

(b) The y-axis of Fig. 3 should either be labeled “absolute number of false positives” or “false positives (%)”. I believe in the present case it doesn’t matter, but the current label forces the reader to go back to the methods section and check how the TCGA dataset was constructed.

Response (7) : (a) This point has been addressed by implementing MAD and Boxplot for both breast cancer and single cell dataset at the highest dimension in figures 3 and 6 respectively. MAD and Boxplot have a much higher false positives % than the robust subspace methods. We added a section 2 (h) to give an overview for the reader on how to implement MAD and Boxplot for outlier detection. The codes to implement this is also added in the GitHub repository.

(b) y-axis of Figures 3 and 6 have been changed to represent false positives %.

Comment(8) : *The author list of ref. (4) is incomplete. Author list of reference [4] has been corrected.*

Responses (8) : Author list of reference [4] has been corrected.

Minor Points:

Comment (5) : *p.2 line 17: techniques to outliers → techniques to determine outliers.*

Response (5) : What was meant in this sentence is machine learning techniques that are robust to outliers (unaffected by outliers). Clarification made in page 2 line 11.

Comment (6) : *p.2 line 31: number of feature increase \rightarrow number of features increases line 31/32: reword “the space the data lives in”. To my understanding, data is not a living object.*

Response (6) : Corrections done in page 2 lines 25-28. Starting at “The problem” ending at “significant results”.

Comment (7) : *p.2 line 51: “gives it much better computational efficiency from the state of the art”.is unclear. Reword. line 52: the sentence “this model same as standard PCA” is incomplete.*

Response (7) : Corrections done in page 2 lines 44-47, from “This convexity” to “local minima”.

Comment (8) *p.3 line 28/29. Sentence “Where the identify ... of the data matrix” is incomplete.*

Response (8) : We meant to write “the identity” other than “the identify”. We have re-worded the sentence to be clearer. Correction made in page 3 lines 21-23, sentence starting “This method...”.

Comment (9) : *p.4 last line: a citation to the ADAM method should be added.*

Response (9) : Page 4 line 30, Reference [29] as been added for ADMM.

Comment (10) : *p.6 line 21: sentence “Which suffers from ... of the domain.” is incomplete*

line 22: will decided \rightarrow will decide

line 23: corresponds \rightarrow correspond

line 23: Sentence “Therefore, giving a cut-off ...” is incomplete.

Response (10) : Corrections made in page 6 lines 14-18.

Comment (11) : *p.8 line 26: by the a author → by the authors line 47: from these figure → from these figures*

Response (11) : Corrections made in caption of Figure 1. Merged Figure 2 (a) and (b) in one figure and added two more figures for PCA and t-SNE (correction of Reviewer 2).

Comment (12) : *p.13 line 45: Sentence “Compared to [26] and [27] ... between samples” is incomplete.*

Response 12 : Corrections made in page 13 line 6 to 7, sentence starting “Other methods ...”.

Response to Reviewer 2

Comment 0 : *The authors should mention clearly that outlier genes can not be detected using this algorithm as the references they chose frequently discussed outlier samples and genes*

Response 0 : This comment has been addressed in page 4 line 30-32, starting at “We also need to ...” ending at “are transposed”. Our method is used to detect outlier samples, but it can be easily amended to detect outlier genes by transposing the matrices of problem 2.2 in manuscript

Comment 1 : *Tuning the value of lambda does not appear to be a straight forward approach. The authors mentioned that the value of lambda “needs to be chosen in such a way that the number of outliers are not too large”. I believe “too large” is too general and can be tumor specific, user specific, case specific, .. etc. Additionally, relying on the long mathematical approach to chose lambda each time a user wants to use the presented algorithm would drive the possible users from biology or pharmacy away. the authors should suggest a simpler way to chose the optimal lambda*

Response 1 : This comment has been addressed in page 6 lines 27-34 and in page 6 line 39-40.

Comment 2 : *it is not clear why the authors used the known outliers in tumor samples in the colon dataset while reference [4] in the manuscript presented the history of the chosen colon dataset and 9 outliers are widely known not only 5. Have the authors tested their presented algorithm on the outliers in normal samples too?*

Response 2 : We have not used the normal samples in the colon cancer dataset. We chose the tumor dataset as that would be of greater interest compared to the normal sample dataset. Correction for this done by better describing the colon cancer dataset from [22] in page 7 lines 34-37, starting at “The 62 samples...” ending at “tumor samples”.

Comment 3 : *choosing the “the most variables genes across sample” is unclear. Why the authors had to discard a huge fraction of the genes? how did the authors decide about how many “variables genes across sample” are needed? For example, in one dataset they chose the top 700 and in another the top 2000.*

Response 3 : The retention of most variable genes is commonly used as a pre-processing step for machine learning algorithms applied to Genomic datasets. Mainly used to reduce noise and filter the most informative genes (most variable ones). We added an explanation of how to choose the number of most variable genes for the three datasets chosen. Correction found in page 8 lines 5-7 sentence starting “The number of ...”, page 8 lines 23-32, page 8 lines 40-42.

Comment 4 : *The authors should mention clearly why they chose to sample TCGA datasets by 100+ and 5- samples and also why exactly 30 datasets. was it a trial-error approach to chose the 100, 5, and 30?*

Response 4 : Corrections made in page 8 lines 17-22, starting at “The choice of ” ending “different samples”.

Comment 5 : *The TCGA dataset has another 429 samples that the authors did not mention. datasets should be described in details even if parts of them will be discarded later*

Response 5 : We used TCGA breast cancer dataset downloaded from UCSC Xena browser. Link to this is in the GitHub repository presented in the manuscript. The dataset has 1218 samples, 600 of which are ER+ and 79 ER- the remaining 439 do not have labels for ER, thus are not used. Correction done in page 8 lines 13 to 14.

Comment 6 : *The single cell measurements can be for gene expression or DNA methylation for example. the users should mention in the datasets section all details about the third dataset.*

Response 6 : Correction done in page 8 lines 34-35, sentence starting “The dataset consists of ...”.

Comment 7 : *the authors should mention if they used RAW or normalized datasets in their testing*

Response 7 : Correction done in page 7 line 32, page 8 line 9-11 sentence starting “Dataset consists...”, page 8 lines 34-36 sentence starting “The dataset consists...”.

Comment 8 *In the results and discussion sections, the authors should stress on the finding that GOP will have less false positives but still will probably miss any outlier that OP might miss.*

Response 8 : We thank the reviewer for this comment. We would also like to clarify that this statement is true only for the colon cancer dataset, but not true for the single cell dataset and breast cancer dataset. For the colon cancer dataset this statements has been added in the manuscript in page 9 lines 16 to 17. For the breast cancer and single cell dataset we add clarification on why the F-score of GOP is higher than OP and other methods, by looking at both the true positives and false positives. This clarifications are done in page 11 lines 4 to 5 sentence starting “In this case...” and page 12 lines 19-21 starting from “In this case” ending “6 outliers”.

Comment 9 : *the authors should mention that basic clustering methods have had similar performance to their presented algorithm. For example, in [4] it is presented that average hierarchical clustering missed only one outlier sample when applied to the same colon dataset the authors used*

Response 9 : This is mentioned in page 9 lines 6-8, sentence starting “It should be noted....”.

Comment 10 : *in Figure 3, what are the 6 boxes referring to? 6 runs for the 30 datasets?*

Response 10 : We apologize for the confusion. We added the description of what each subdivision of Figure 3 is. Each subdivision represents running GOP, OP and Gaussian density method on the 30 datasets on a specific dimension. Correction added in page 11 Figure 3 caption.

Comment 11 : *The authors mentioned that “the performance of the robust subspace methods improves with the increase in dimensionality, ” how can it avoid falling in the curse of dimensionality?*

Response 11 : We added explanation on why the robust subspace methods do not fall in the curse of dimensionality in page 12 lines 6-9 ,and page 14 lines 3-4 sentence starting “Thus finding....”.

Comment 12 : *Consistency issues:*

A. OP and GOP results always appeared in the same section except in the analysis of the colon dataset they were separated in two sections.

B. PCA and t-SNE were tested on the TCGA dataset only

C. in the datasets section, the number of chosen “most variables genes” was mentioned in the colon dataset section but for other datasets it was mentioned in the results. It should be in the same place for all of the datasets

Response 12 : A. We merged results of GOP and OP in same section for colon cancer dataset, this is shown in section 3(a).

B. We would like to clarify that PCA and t-SNE have been tested in TCGA and single cell dataset in the original manuscript. We added PCA and t-SNE testing for the colon cancer dataset, Figure 2 in revised manuscript.

C. We addressed this comment by mentioning the number of chosen most variable genes in section 2(i) (Datasets and Preparation). We made modifications in page 8 lines 29-30 Sentence starting “For each of the 30” and page 8 lines 40-42 sentence starting ‘For each of the 30 constructed’.

Comment 13 : *In the references, reference [4] is missing one author name. I suggest that the authors double check the whole reference list for missing authors in other references. Additionally,*

Response 13 : Reference list corrected.